# Investigation of the Internal Structure of Hardened 3D-Printed Concrete by X-CT Scanning and Its Influence on the Mechanical Performance

**DOI:** 10.3390/ma16062534

**Published:** 2023-03-22

**Authors:** Yanjuan Chen, Jukka Kuva, Ashish Mohite, Zhongsen Li, Hubert Rahier, Fahim Al-Neshawy, Jiangpeng Shu

**Affiliations:** 1Department Materials and Chemistry, Vrije Universiteit Brussel, Pleinlaan 2, 1050 Brussels, Belgium; 2Circular Economy Solutions, Geological Survey of Finland, 02150 Espoo, Finland; 3Department of Mechanical Engineering, Aalto University, 02150 Espoo, Finland; 4Hyperion Robotics Oy, 02630 Espoo, Finland; 5Department of Civil Engineering, Aalto University, 02150 Espoo, Finland; 6Department of Civil and Architectural Engineering, Zhejiang University, Hangzhou 310058, China

**Keywords:** 3D-printed concrete, hardened concrete, interface, internal structure, X-ray computed tomography

## Abstract

As we know, 3DPC is printed layer by layer compared with mold-casting conventional concrete. Pore structure and layer-to-layer interface are two main aspects of the internal structure for 3DPC, which decide 3DPC’s mechanical performance. The layer-to-layer interface caused by printing is specific to 3DPC. The emphasis of this study lies in the layer-to-layer interfaces of 3DPC. The first aim of this study is to quantify the characteristics of the layer-to-layer interface and therefore characterize different aspects of the interfaces. The second aim of this study is to explore how the internal structure of printed concrete influences the mechanical performance of 3DPC. This research set out to design a series of experimental comparisons between 3DPC and casted concrete with the same compositions. Mechanical tests, i.e., compressive stress, ultrasonic Pulse Velocity test, flexural tension, and tension splitting, as well as the Ultrasonic Pulse Velocity test, were performed to check the mechanical performance of 3DPC. Contrary to what has often been expected, the mechanical test results showed the printed concrete has a quality not worse than casted concrete with the same recipe. Meanwhile, the X-ray computed tomography (X-CT) is used to characterize the internal structure, pore shapes, and interfaces of 3DPC. First, the investigation revealed that the lower total porosity and fewer big voids could be the fundamental causes meaning 3DPC has a better mechanical performance than casted concrete. Second, the statistics based on aspect ratio show that the distribution curves follow similar trends, regardless of the printed or casted concrete. Third, this study quantified the depth of the different interfaces for 3DPC. The results suggest that the porosity in an interface varies in a range. The author’s pioneer work has contributed to our present understanding of the interfaces of 3DPC.

## 1. Introduction

The application of 3D concrete printing (3DCP) in the construction industry has gained significant attention because of the considerable benefits over conventional construction methods. Such benefits include improved geometrical freedom, greater safety in construction, formwork and mold-free manufacturing, and reduction in construction time, labor, cost, and waste. It has been shown that 3DCP reduces CO_2_ emissions by about 86% and energy consumption by about 87% compared to a precast counterpart according to the life cycle analysis [1]. Compared with mold-casting conventional concrete, 3DPC is printed layer by layer. Current structural and durability design codes consider concrete as a homogeneous material. For 3D-printed concrete elements, this is no longer the case, due to the layered concept of 3D-printed elements, in which layer-to-layer interfaces are unavoidable. 

The layered nature of 3DPC might result in more porous and weaker interfaces, which will lead to anisotropy in mechanical properties, weaken the overall load-bearing capacities, and influence the transport and durability behaviors of 3DPC structures [2,3,4,5]. Thus, regarding a 3DPC structure or element, the role of layered interfaces is significant. Structure design models will have to consider this layered nature of the structure, which will most probably no longer be provided in the traditional way.

Owing to this significance, the research interest on layer-to-layer interfaces of 3DPC is growing [6,7,8]. The research is mainly focused on two aspects: (1) enhancing the bond strength between layers [9,10,11,12,13,14,15,16]; (2) anisotropy in mechanical properties resulting from layer-to-layer interfaces [17,18,19,20,21,22,23]. Apart from the two aspects, several researchers also studied how printing process parameters, e.g., depositing height, layers time intervals [24,25,26,27], influence the bonding strength between interfaces. However, with respect to aspect (1), the enhancing methods either increase the cost of printing concrete or make the automated printing process complicated and might affect the shape stability of the bottom layers. With respect aspect (2), the mechanical results of one study may not be in line with the others due to the variability in printing processes and test procedures. This is not only need to unify the test standard approach to test mechanical properties, i.e., bonding strength but also to profoundly understand the layer-to-layer interfaces. 

Since many issues resulting from the interfaces of 3DPC are still open and need to be investigated sufficiently, several studies to date have carried on the interface analysis of 3D-printed mortar. For instance, Liu et al. [28] and Chen et al. [29] have evaluated the width of the interface area and the average of porosity with X-ray computed tomography (CT) scanning quantitively, separately. 

So far, however, there is less information about the specific definition of the printing layered interfaces, and there has been very little discussion about the characteristics of the interfaces. For instance, with respect to the number of interfaces, most of the research only showed several layers in the 3D CT rendering, which was limited to four layers (three interfaces) [20,25,28]. Xin Huang et al. have presented five interlayers in 3D CT reconstruction [27]. It probably is limited by the suitable sampling size or buildability of printed concrete. The maximum number of layers printed before collapsing is essential and has been used as an indicator of buildability in the existing literature [30,31,32]. Therefore, the sampling way for printed concrete should be defined. A profound understanding of the characteristics of layer-to-layer interfaces is a prerequisite for enhancing bonding strength and studying mechanical properties, transport, and durability behavior of 3DPC [33,34]. In particular, quantification of the characteristics of the layered interfaces is critical. 

Therefore, the emphasis of our study lies in the layer-to-layer interfaces of 3DPC. More interlayers of printed concrete are presented in 3D X-CT reconstruction compared to the existing literature. The aim of this study is twofold. The first is to quantify the characteristics of the layer-to-layer interface and therefore characterize different aspects of the interfaces. The second is to elaborate on how the internal structure of printed concrete influences the mechanical performance of 3DPC.

## 2. Materials and Methods

### 2.1. Materials

In this research, a custom-designed printable mixture was adopted, containing Portland cement (CEM I 52.5 R) blended silica fume, granite aggregates with a maximum particle size of 2 mm, limestone filler, calcium formate as setting time modifier, PCE (polycarboxylate ether) dispersants as rheology control additives, and a small amount of PVA fibers. The water-to-binder ratio was 0.22 by volume, or 0.36 by weight. 

### 2.2. Printing Process and Mold-Casting Process

The printed concrete slab under investigation is contributed by Hyperion Oy in Finland without any commercial purpose, aiming to contribute to modern concrete research.

The slab is printed with a 3DPC setup as shown in Figure 1. The setup consists of a degree-of-freedom (DOF) gantry robot and a M-Tec Duomix 2000 mixer-pump (made by m-tec, Neuenburg, Germany) with a linear displacement pump, which feeds concrete through a 25 mm diameter nozzle connected to a 10 m length hose. The printed slab consists of the path as shown in Figure 2. After printing, the slab was covered with a plastic sheet for 24 h. At an age of 50 days, it was removed for saw-cut to obtain investigated specimens. 

The same mixture was cast in several steel molds of 10 cm × 10 cm × 10 cm and compacted with a vibrating table. Immediately after casting, the samples were covered with a plastic sheet to prevent moisture loss and left to harden at room temperature (20 °C). After 24 h, the samples were demolded and cured in the standard curing room at 100% RH and 20 °C for 50 days. 

### 2.3. Sampling

In this research, the specimens under investigation were manufactured from both printed and mold-cast states, with the purpose of comparing the differences in internal structure of concretes and mechanical performances between printed and mold-cast states. The sampling process is presented in Figure 3. There are two types of interfaces, one is perpendicular to the z-axis, which will be affected by the gravity. These are marked as vertical interfaces (V-interface). The other is perpendicular to the y-axis. These are marked as horizontal interfaces (H-interface). The V-type and H-type interfaces intersect in the whole slab. The thickness of each layer perpendicular to the z-axis is around 12–15 mm, the width of layers perpendicular to the y-axis is around 30 mm. The loading usually comes from three different directions (x, y, z), and it will relate to the interfaces [6]; the specimen with the size of 5 cm × 5 cm × 10 cm is adopted to investigate the mechanical performance because specimen with this size will cover both types of interfaces. Thus, printed concrete specimens with size of 5 cm × 5 cm × 10 cm were sawn from the printed slab from the three different directions (x, y, and z-axis) as shown in Figure 4. For printed specimens, the longitudinal axes of specimens along the x, y, and z-axis are denoted x-orientation, y-orientation, and z-orientation, respectively (see Figure 5). 

In the meanwhile, casted specimens of the same size were sawn from the 10 cm cubic mold-cast specimens.

An experimental program has been designed to investigate the mechanical properties of 3D-printed specimens with different interfaces’ distributions and the casted specimen using destructive mechanical tests and non-destructive tests, namely, compressive strength, flexural tension, tensile splitting, and Ultrasonic Pulse Velocity (UPV). 

### 2.4. Compressive Strength Tests

Compressive strength was measured in prismatic specimens from both 3D-printed and mold-cast specimens, according to the SFS-EN 12390-3 standard. The prismatic specimens were 5 × 5 × 10 cm and were loaded with a rate of 0.6 N/mm² s until failure. The maximum load $F_{max}$ was recorded to calculate the compressive strength $f_{c}$ in MPa ($N/{\mathrm{mm}}^{2}$) as follows:(1)$$f_{c}=\frac{F_{max}}{A_{c}}$$
where $A_{c}$ is the cross-sectional area of the specimen, in ${\mathrm{mm}}^{2}$, which might vary around 2500 ${\mathrm{mm}}^{2}$ due to errors from sample sawing. 

### 2.5. Flexural Tension Tests

Three-point bending test were performed on the prismatic specimens in accordance with the SFS EN 196-1 standard. A load was applied on the specimens vertically by means of the loading roller to opposite side face of the prism at a rate of 50N/s until fracture. The flexural strength $R_{f}$ in MPa was calculated from:(2)$$R_{f}=\frac{1.5\times F_{f}\times l}{b^{3}}$$
where b is the side of the square section of the specimen, in mm, which is around 50 mm*;* $F_{f}$ is the load applied to the middle of the specimen at fracture, in N; and l is the distance between the supports, in mm, here equal to 80 mm. 

### 2.6. Tensile Splitting Tests

Tensile splitting strength test were performed on prismatic specimens, according to the SFS/EN 12390-6 standard. The initial load applied on specimens did not exceed approximately 20% of the failure load. Afterwards, load was applied to the specimen without shock and increased with a constant rate of 390 N/s continuously. At failure, the maximum load was recorded. The tensile splitting strength was calculated as:(3)$$f_{ct}=\frac{2\times F}{\pi \times L\times d}$$
where $f_{ct}$ is the tensile splitting strength, in MPa $\left(N/{\mathrm{mm}}^{2}\right)$; F is the maximum load, in N; L is the length of the specimen, in mm; and *d* is the designated cross-sectional dimension, in mm, here around 50 mm. 

### 2.7. X-CT Test

X-ray computed tomography (X-CT) is a powerful and non-destructive tool to create 3D visualizations of the internal structure of cementitious materials. The 3D data can be used for qualitative and quantitative investigation [35,36,37,38]. For example, the chemical composition and density of materials can be indicated by the gray value or contrast of images in X-CT tomographs [39,40]. The grayscale value of these images can reflect the density of the different area of the concrete. Therefore, a certain gray value as a threshold can be set to segment different part of the image. In this way, the different parts for printed concrete, i.e., pore, matrix, and interface, can be segmented. The quantitative parameters, e.g., porosity, can be calculated based on the grayscale value. We process the image analysis and render the 3D inter-structure with the software of PerGeos version number: 2020.2. Moreover, we conducted the statistical analysis of 3D aspect ratio and performed relationship between elongation, flatness, and volume of voids with bubble figures by coding with Python 3.9 in Section 4.2. 

## 3. Results and Discussion

### 3.1. Compressive Strength and Ultrasonic Pulse Velocity (UPV) Test Results

Figure 6 shows the fracture patterns after loading. It can be observed that x-orientation specimens have fewer interfaces compared with the other two orientations. Therefore, it is not easy to fracture along the loading direction. As for the z-orientation specimens, there are more interfaces (around four V-interfaces) along the loading direction compared with y-orientation specimens (only one H-interface); thus, z-orientation specimens probably are stronger than y-orientation samples because interfaces are the weak joint in the printed concrete. 

From the compression test results shown in Figure 7, it was observed that the x-orientation is stronger than the other two orientations, as well as the casted one. On average, x-orientation was 32%, 20%, and 50% stronger than y-orientation, z-orientation, and casted specimens, respectively. The orientational dependency in the compressive strength were obvious. The remarkable variations regarding compressive strength in different orientations for the printed specimens are due to the different interlayers’ distribution in the loading, in particular the interfaces distribution along the loading direction. Moreover, it is surprising that the compressive strength value for x-orientation is higher than the casted one. Subsequently, the internal structure part investigated by X-CT will offer a reasonable explanation. 

Meanwhile, the Ultrasonic Pulse Velocity test (UPV) was conducted on the paralleled specimens to investigate the quality of concrete. High pulse velocity values generally indicate a good quality of concrete. The Ultrasonic Pulse Velocity device “Matest Ultrasonic Pulse Velocity Tester C373N” was calibrated using the calibration reference bar before conducting each measurement. The measurements were performed using the UPV indirect transmission according to the “SFS-EN12504-4,2004” standard. Qualities of concrete for different values of UPV are presented in Table 1. 

Based on the previous research from Fahim et al. [42], the compressive strength (MPa) tested according to the SFS-EN12504-1, 2019 standard has a better correlation with the UPV pulse velocity (km/s), as shown in Equation (4):(4)$$\sigma_{c}=23.76e^{0.214v}$$
where $\sigma_{c}$ is the correlated compressive strength, MPa, and *v* is the UPV velocity (km/s). It can be seen from the equation that a higher value of UPV indicated a higher value of compressive strength. In this research, the test results for the UPV and the correlated compressive strength are presented in Table 2. 

Combined with the information in Table 1 and the results in Table 2, it can be estimated that x-orientation specimens have a good quality, while the casted specimen has a poor quality compared to other orientations. This can confirm that the casted specimen has a poor compressive strength compared to the printed ones to some extent. Therefore, the UPV test results basically support the conclusion obtained from compressive strength test results, despite some researchers reporting no significant difference in compressive strength between casted and printed samples [43]. Concerning the comparisons between printed concrete and casted concrete regarding the compressive strength, the conclusions might vary from one study to another. In the authors’ opinion, it could be mainly attributed to the way of sampling, and therefore, specimens in each orientation might encounter different interlayers. Thus, regardless the samples from different orientation of printed concrete or casted concrete, the remarkable variations are due to the different interlayers’ distribution in the loading, in particular the interfaces distribution along the loading direction.

### 3.2. Flexural Strength Results

The failure patterns for specimens were observed (in Figure 8) when bending is applied. The failure patterns for printed specimens are inclined to both sides in comparison with failure patterns for casted specimens located in the center, owing to the different interlayer distribution across the specimens. The interlayers, as the weaker joints of concrete, can transfer the stress. Figure 9 shows a summary of flexural results of specimens from different orientations. The flexural results follow the reduction trend in compressive strength results (see the black line in Figure 7 and Figure 9). There is an average increase of 9.7%, 4.9%, and 32% for x-orientation specimens compared to y-orientation, z-orientation, and casted specimens, respectively. 

This is attributed to the interlayer distribution in the loading direction. As we know, interlayers are the weak joints for 3D-printed concrete. For x-orientation specimens, loading is parallel to the V-interfaces (as shown in the third column of Figure 8) and more V-interfaces (estimated 4–5 V-interfaces) exist in these specimens; in this case, the tensile force (as depicted in Figure 10) from the bending is not required to overcome the interfacial bonding strength. In contrast to y- and z-orientation specimens, tensile force from bending must overcome the interfacial bonding strength from the H-interfaces, and interfacial bonding strength from V-interfaces (as shown in the third column of Figure 8), respectively. In this 3DPC scenario, the H-interface (horizonal) and V-interface (vertical) are two different types of interfacial bonding due to the different interval printing time and printing way. The V-interface could be bonded more tightly in comparison with the H-interface. On the one hand, it is due to a mechanical force during printing and the gravity in the vertical orientation. On the other hand, the interval printing time for the V-interface is less than the H-interface, which will result in the interfacial bond strength for V-interface being greater than H-interfaces. Wolfs et al. [6] have proved that interlayer bonding strength reduces as the interlayer time interval increases. Consequently, the tensile force from bending will be easier to overcome in the H-interfaces, in comparison to the V-interfaces. Thus, this can explain z-orientation specimens exhibit better flexural resistance than y-orientation specimens as shown in Figure 9. 

### 3.3. Splitting Tension Results

It can be seen that the 3DPC specimens fractured into pieces along with the interfaces under the loading of splitting tension, and the interlayers’ boundaries are clear on every fractured section, as shown in the second column of Figure 11. However, as depicted in Figure 12, in terms of orientations, the tensile splitting results do not follow this reduction trend as compared to the results of the compressive test and flexural test. By contrast, there was an average decrease of 37% and 32% for x-orientation specimens, respectively, compared to the other two orientations’ (y, z) specimens. It is indicated that the splitting tension resistance for 3DPC significantly tends to interlayers’ orientation dependency. 

Strips were used to transfer the load on the specimen when conducting the splitting tensile strength test on a 150 mm prism, the force diagram was shown in Figure 13. Basically, there are two aspects of 3DPC that contribute to resisting the lateral tensile stress: (1) cohesion, due to the contribution of binder adhesion and aggregate interlocking; (2) interlayers bonding, due to the bonding strength at concrete-to-concrete interfaces between different printed layers at different times. Apparently, the lateral tensile stress resistance should be variable because it depends on the weakest part of the concrete. The layer-to-layer interface is the weakest joints of printed concrete; therefore, the lateral tensile strength must overcome the interfacial bonding strength to reach failure. Thus, the splitting tensile strength values should correlate to the interlayers bonding strength. As the third column of Figure 11 depicted, for x-orientation specimens, the lateral tensile stress from the loading must overcome interfacial bonding strength from several V-interfaces (vertical) due to loading is parallel to the V-interfaces. In contrast to y- and z-orientation specimens, the lateral tensile stress from the loading do not need to overcome any interfacial bonding stress. Therefore, the x-orientation specimens have the poorest splitting tensile resistance by comparison to the other two orientations, due to the interfaces being the weakest joint of printed concrete. As shown in the third column of Figure 11 and Figure 13, for x-orientation specimens, the lateral tensile stress from the loading is parallel to bonding force from V-interfaces, which mainly contribute to overcoming the interfacial bond strength from V-interfaces; for y-orientation and z-orientation specimens, the lateral tensile stress from the loading is parallel to H-interfaces and V-interfaces, respectively. Consequently, in the two cases, the lateral tensile stress almost no need to overcome any interfacial bond strength but it needs to overcome the binder adhesion and aggregate interlocking; in the meanwhile, the gaps of interfaces can promote propagation of the lateral tensile stress and relieve the internal pressure of printed concrete. Thus, this can explain that splitting tensile strength for x-orientation specimens is significantly less than y-orientation and x-orientation specimens. As for the slightly difference in values of splitting tensile strength for y- and z-orientation specimens, it might be due to the different types of interfaces, which have different bonding strengths.

In conclusion, the mechanical performance of printed concrete significantly tends to interlayers’ orientation dependency, especially the splitting tensile resistance. It depends on the interlayer distribution across the printed concrete, the types of layer-to-layer interfaces, and even the number of interfaces. This is because the layer-to-layer interfaces are the weakest joints in printed concrete, and they greatly affect the integrity of printed concrete. During the application of printed concrete as a load-bearing structure, more attention should be paid to interlayer’s orientation of the printed concrete member, as it is a basis for safe structure design. Furthermore, the interfacial bonding strength is critical for evaluation of mechanical performance of printed concrete. However, there is no standard approach to test the interfacial bonding of 3D-printed concrete. Various studies followed different approaches, and the finding of one study may not transfer to others due to the different the differences in printing processes and test procedures. Thus, it is necessary to define a unified standard to evaluate the interfacial bonding strength of 3D-printed concrete, including defining the types of tests for interfacial strength, specimen size, loading direction, etc. in the 3DPC field.

## 4. X-CT Results Analysis

In principle, X-ray computed tomography (X-CT) is a 3D imaging technique which is based on the 3D-computed reconstruction of a sample from the 2D radiographic projections acquired from different angles around its axis of rotation. 

### 4.1. Samples, Equipment, and Method

In this study, three 3D-printed concrete samples with different printing orientations and one casted sample as the reference were scanned to determine how printing orientation affects their internal structure. Specifically, three cores of 40 mm diameter and 100 mm height (Figure 14) were drilled from the three different orientations of 3D-printed concrete, while the same size core was drilled from casted concrete as a reference. These samples were scanned using the 240 kV microfocus tube with 0.5 mm of copper as a beam filter. Accelerating voltage was 150 kV and tube current 180 μA for a total power of 27 W. At each angle, the detector waited for a single exposure time of 500 ms and then took an average over three exposures of 500 ms. The image resolution was 25.05 μm. Samples were imaged with a helical scan, where the sample moves vertically while it rotates. Depending on the scan, there were 6239–6318 angles for a total rotation angle of 1110–1124° and a total scan time of 208–211 min. No ring artifact reduction was used for any of the samples in reconstruction, and the beam hardening correction coefficient (0–10) was 5. The X-CT test work was carried out in the laboratory of Circular Economy Solutions unit of the Geological Survey of Finland (Geologian tutkimuskeskus). 

### 4.2. Interfaces Characterizing for 3DPC from X-CT Results

As per the 3D rending displayed in Figure 15, the 3DPC samples exhibit clear printing interfaces distributed across the whole samples compared with the casted sample. In particular, as shown in the image of the 3DPC-Z (with perspective), it can be seen that the H-interface exhibits a zigzag shape, which is apparently wider than the V-interface. 

Figure 16 shows the average grayscale value of the sample along the longitudinal axis, which can be interpreted as relative density. The greyscale values reveal a significant difference in casted concrete (black) and 3DPC (blue, yellow, and red). Moreover, the greyscale values curves for different orientations of 3DPC are close to each other by comparison with the casted one. In detail, the trends of curves for different orientations of 3DPC vary due to the interfaces existing. There are zigzag peaks in curves for 3DPC-Y and 3DPC-Z. Regarding 3DPC-Y (the yellow), it can be seen there are three wide peaks (upwards) overall. The distance of the highest points for the two adjacent peaks is approximately 30 mm, coincidently, which is equivalent to the thickness of each layer (30 mm) along the longitudinal axis as shown in Figure 15. Similarly, regarding the 3DPC-Z (the red), there are eight sharp peaks (upwards) overall. The distance for each pair of adjacent peaks is around 12–15 mm, equivalent to the thickness of each layer (about 12–15 mm) along the longitudinal axis. Together, the present findings confirm these positions that peaks emerge are located at printing interfaces. Moreover, for these peaks on the greyscale values curve of 3DPC-Z, it exhibited significant deviations. This is due to the difference in the density of different printing interfaces. Further, there are no printing interfaces across the longitudinal direction of the 3DPC-X sample; correspondingly, there are no significant peaks in the curve of 3DPC-X (the orange). In addition, the peaks in the curve of the casted concrete (the black) are not as regular as in the 3DPCs. These indicated that the greyscale values for each peak might tell the printing interface from the matrix. Thus, the result of this analysis clearly supports that greyscale values can quantity and characterize the matrix and printing interfaces of 3DPC.

### 4.3. Pore Information Characterizing for 3DPC from the X-CT Results

#### 4.3.1. Pores’ Shape Characterizing

As shown in Figure 17, the pores in the casted samples exhibit circular shapes, while the pores in the 3DPC samples mostly exhibit irregular shapes (approximately ellipsoidal shapes). It agrees with some findings from other researchers regarding the pore shape observation based on 2D images [20,28]. Meanwhile, the size of pores with a good representativeness are marked in Figure 17. The observed pores in Figure 17 are categorized to air voids according to the definition of air void; it can be observed that the casted sample apparently has larger pore volume with comparison to 3DPC samples. This is consistent with Section 4.3.4 (it will be mentioned later), suggests that the casted sample has more voids with an equivalent diameter of more than 80 μm by comparison with 3DPCs. If there are more regular (circular) pores in casted concrete than printed concrete, as observed in Figure 17, this was probably because the casted samples were vibrated during preparation, while the 3DPC samples were not. The vibration during concrete preparation helps form spherical air voids. However, the visual estimation of the voids’ shape is not sufficient to come to a conclusion. Therefore, we calculated the elongation index (EI) and flatness index (FI) of voids for all concrete aiming at describing the shape of voids.

Moreover, a 3D rending of several representative single irregular pores for 3DPC were visualized as Figure 18 shown. The irregular shapes could be described as prolate, oblate, bladed, etc. It means the irregular pores with these shapes do exist in the concrete. Further, the amount/probability of these irregular shapes will be elaborated in Figure 19 and Figure 20. 

For the purpose of investigating the voids shapes in concrete further, we statistically analyzed the aspect ratio of voids in 3DPC concretes, as well as in the casted concrete. The aspect ratio (AR^3D^) of a 3D void, as a parameter of shape descriptors, which quantify the form of a pore/void, is defined as the average value of elongation index (EI) and flatness index (FI) [44,45]. Elongation is the ratio between major and intermediate axes of the ellipsoid to a void; flatness is the ratio between intermediate and minor axes of the ellipsoid to a void [46]. Figure 19 shows the probability distributions of 3D voids’ aspect ratio for concrete. It can be seen that the statistic bars show similar trends regarding the aspect ratio, regardless of the printed concretes or casted concrete. The voids with an aspect ratio of 0.74–0.82 take up most of all voids, more than 55%. 

Meanwhile, aiming to explore the difference further between printed concrete and casted concrete, we visualized the relationship between elongation, flatness, and volume (bubble size) of a particular void in relation to the void shapes with the bubble figure [46], as shown in Figure 20. It suggests the information as mentioned below. First, basically, the voids in the X sample and Z sample printed concretes tend to be more prolate by comparing casted concrete; however, voids in the Y sample are as equant as the casted sample. This could be explained by the fact that the areas of interfaces for the X and Z samples are greater than those for the Y sample. Namely, the bigger the area of interfaces the printed sample has, the more prolate voids it would have. Second, the big voids are more equant compared to small voids for casted concrete. Last, for the printed concrete, the Z sample is the best by comparison to the X and Y samples regarding the void shapes, because we expect the voids in concrete to be as equant as possible. In addition, the bubble figure also suggests that the casted concrete has more big voids than the printed concrete. The big voids can significantly reduce the strength of concrete: the more big voids there are, the weaker the concrete. 

#### 4.3.2. Profile of the Porosity Regarding the Height of Different Samples

As shown in Figure 21, the curves for the porosity profile trends inversely resemble the grayscale values curves (as shown in Figure 16). It can be observed that the values of porosity fluctuated regarding the height of samples. The porosity profile for 3DPC-X is the least fluctuated by comparison with the others. The pore information of 3DPCs and casted concrete were listed in Table 3. It was presented that the casted sample is greater than 3DPCs regarding the average of total porosity. This suggests that the casted sample might have poorer strength than 3DPCs. Thus, the results of porosity support the conclusion drawn in Section 3.1. However, as for the 3DPCs, the average values of total porosity deviate in a reasonable range (4.5±0.8%) in different orientations, which indicates that 3DPCs are especially heterogeneous. Whilst the values are not positively correlated with the strength values (see Section 3.1), there are two reasons behind this. For one thing, apart from the total porosity, the interfaces affect the strength as well. For another, the strength resistance of the 3DPCs depends on the weakest part of the specimen. The weakest part might exist in either in interfaces area or intersection area of the different interfaces. Therefore, the findings on total porosity suggest that we cannot estimate the strength of 3DPCs solely based on the total porosity because the strength resistance of 3DPCs depends not only on the total porosity but also on the weakest area of the interfaces of specimens. 

#### 4.3.3. Estimate the Depth of Printing Interfaces

We mentioned above that the porosity profile trends inversely resemble the grayscale values curves. Then, as for 3DPC-Z, the significant sharp peaks (downwards) in Figure 21 would be in printing interfaces according to the analysis from Section 4.2. If we define the area that suffers the sharpest transition area for porosity as the layer-by-layer interfaces, the distance between the adjacent valley and peak would be the interface’s thickness. Taking 3DPC-Z and 3DPC-Y, for instance, we statistically analyze the distance for adjacent valley and peak, respectively, and this is summarized in Table 4. As mentioned in Section 3.1, there are mainly horizontal interfaces (H-interface) in the 3DPC-Y samples and vertical interfaces (V-interface) in the 3DPC-Z samples. Then, the depth for H-interface and V-interface might be around 14 mm and 2 mm, respectively. The shorter printing interval time and the gravity in the vertical orientation result in the depth of the V-interface being less than the H-interface. Further, based on the afore-defined printing interfaces, it should be noted that they are less porous in part of the interfaces area for 3DPC-Z (see the sharp valley in Figure 21). On the contrary, we usually expected the printing interfaces to be a more porous area. Apparently, it is not consistent with what we expected. However, the printing interfaces are a contiguous area of layer upon layer; the lower layer will be more compact than the upper layer due to the gravity in the vertical direction and the mechanical force during printing. Since there was a follow-up operation to trim the matrix’s (layer’s) face after printing a layer. Therefore, the lower layer will be less porous than the upper layer, and then, it is reasonable that the profile of the 3D porosity curve shows sharp adjacent valleys and peaks in Figure 21. 

Further, the porosity for the interfaces can be observed from Figure 21. The sharply adjacent valley and peak regularly emerged on the porosity profile. The data of porosity for interfaces are presented in Table 5. Basically, the minimum values of porosity for the V-interfaces tend to be lower, while the maximum values are greater by comparison with H-interfaces; namely, the porosity values for V-interfaces have a greater range compared to H-interfaces. However, the data for porosity are not sufficient to suggest the bonding strength of both types of interfaces despite the fact that the interfaces’ porosity should be considered as the key point when investigating the bonding strength of the interfaces. Apart from the porosity of interfaces, the other factors, i.e., the depth/width, pore distribution, and even pores’ shape, also are worth considering. Unfortunately, we did not examine the bonding strength of the interface for specimens in this study. Further research should be performed to investigate how these factors affect the bonding strength of the printing interfaces. Meanwhile, further study on defining and unifying the testing standard of the printing interfaces are therefore recommended. 

Moreover, specimens’ mechanical strength (including compressive, flexural, and tension splitting) can be solely attributed to neither the bonding strength of interfaces nor the total porosity of specimens, because the force diagram of specimens under the loading (as shown in Section 3) still be needed to consider as well.

In conclusion, three key points can be considered when evaluating the mechanical strength of 3DPCs: first, the total porosity of the specimen; second, the weakest area of the specimen, and of course, the weakest area can be estimated based on the corresponding pore information; third, evaluate the weakest area suffering loads or not with a force diagram when the specimen is under loading.

#### 4.3.4. Pore Size Distribution of Different Concrete

In this study, the pore volume was detected by X-CT reconstruction technology. However, we converted a pore’s volume to the equivalent sphere diameter when performing the pore size distribution statistic. We emphasized that the diameter of the smallest pore detected by X-CT is more than 25.05 μm due to the image resolution of X-CT scanning being 25.05 μm. Thus, the statistics of the pore diameter range begin from 20 μm, and the step is 20 μm, covering 400 steps in total when conducting the pore size distribution statistic. Namely, the curve in Figure 22 originates from 20 μm corresponding to the X-axis, Typically, the equivalent diameter of pores more than 20 μm was categorized into air voids, and air voids assumed no connection with one another. Therefore, pore connectivity was no longer considered in this study due to the limit of the image resolution.

Figure 22a,b show the trends of the pore size distribution of concretes from X-CT scanning. It was found that 3DPCs can be clearly distinguished from casted concrete. The pore size distribution curves for the samples of 3DPCs are consistent, especially in Figure 22b, even though the samples of 3DPC come from different locations and have different interfaces orientations. Moreover, Figure 22 shows that the peaks of the pore size distribution curve are located at 60 μm according to X-axis except for the originated points, regardless of the 3DPCs or cast concrete. It suggests the pores in the range of 80–100 μm account for most of the total number of pores in the concretes. The 3DPC has a smaller ratio regarding the pores in the range of 80–300 μm and a more significant ratio regarding the pores in the range of 20–80 μm as compared to casted concrete, respectively, as shown in Figure 22b. It indicates 3DPC has more fine pores less than 80 μm in diameter than casted concrete. The gained information here agrees with the bubble figure: casted concrete has more big voids than printed concretes. Coarse pores (big voids) have a significant effect on the strength of concrete, namely, the greater the amount of coarser pores, the greater the reduction in the strength of the concrete. On the contrary, fine pores have no significant effect on the strength of concrete. This can explain why 3DPC has a greater strength than casted concrete with the same recipe, as shown in Section 3.

## 5. Conclusions

This research was carried out to design a series of experimental comparisons between 3DPC and casted concrete with the same compositions. The first aim of this study was to quantify the characteristics of the layer-to-layer interface of 3DPC. The second aim of this study was to investigate how the internal structure of 3DPC affects the mechanical strength of 3DPC. The findings reported here will enhance our understanding of the printing interfaces and the mechanical performances of 3DPC.

(1)The results of mechanical investigation (including compressive stress, ultrasonic Pulse Velocity test, flexural tension, and tensile splitting) have shown that 3DPC significantly tends toward interlayers’ orientation dependency with respect to mechanical performance, especially regarding the splitting tensile resistance. The results also showed that 3DPC has good mechanical performance, i.e., compressive strength, flexural strength, and tensile splitting strength, which is not inferior to casted concrete with the same recipe.(2)The results from X-CT revealed that higher total porosity and more coarse voids in casted concrete than 3DPC could be the fundamental reason for its slightly inferior strength. The results of this study indicate that specimens’ mechanical strength can be solely attributed to neither the bonding strength of interfaces nor the total porosity of specimens; still, the force diagram of specimens under loading (as shown in Section 3) requires consideration due to its interlayers’ orientation dependency.(3)Concerning the X-CT analysis, the first major finding was that porosity profiles for 3DPC (samples with several interfaces, i.e., 3DPC-Y, Z) showed several regular adjacent valleys and peaks with comparison to casted concrete, and that the distance between the adjacent valley and peak happens to be equal to the thickness of each layer of matrix. Based on this finding, this study first attempted to provide the specific definition of the layer-to-layer interfaces and therefore quantified the depth/width of the interfaces. Moreover, the porosity results suggested that porosity in an interface area varies in a certain range.(4)The second significant finding was that the distribution curves regarding the aspect ratio statistic follow similar trends regardless of the type of concrete (printed or casted), and coarser voids in casted concrete are somewhat more “equant” compared to printed concrete.(5)The third significant finding was the samples from different orientations of 3DPC had a unified pore size distribution, which can be easily distinguished from the casted concrete.

The study is limited by the absence of bonding strength tests due to a lack of unified and mature testing standards. Whether or not bonding strength is the key point that affects the mechanical performance of 3DPC might depend on the stress conditions from loading. Further research is needed to investigate how these factors, i.e., numbers of interlayers, the porosity of interfaces, the depth/width of interfaces, pore distribution and pores shape in the interface area, affect the bonding strength of the printing interfaces, as well as the transport properties of printed concrete. Meanwhile, further study on defining and unifying the testing standard of the printing interfaces is therefore recommended.

## Figures and Tables

**Figure 1 materials-16-02534-f001:**
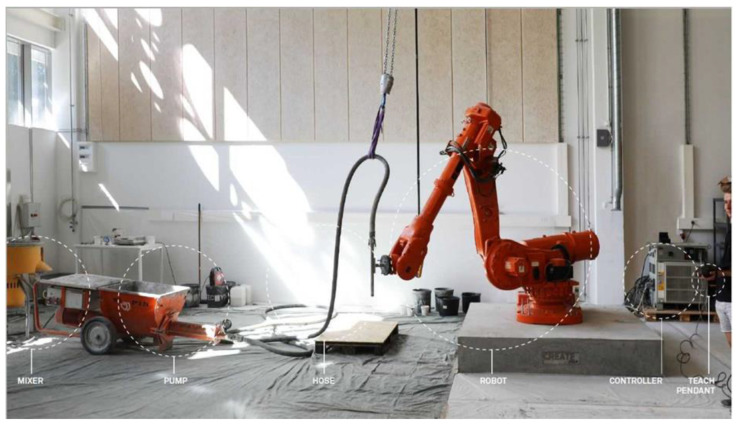
3D concrete printer.

**Figure 2 materials-16-02534-f002:**
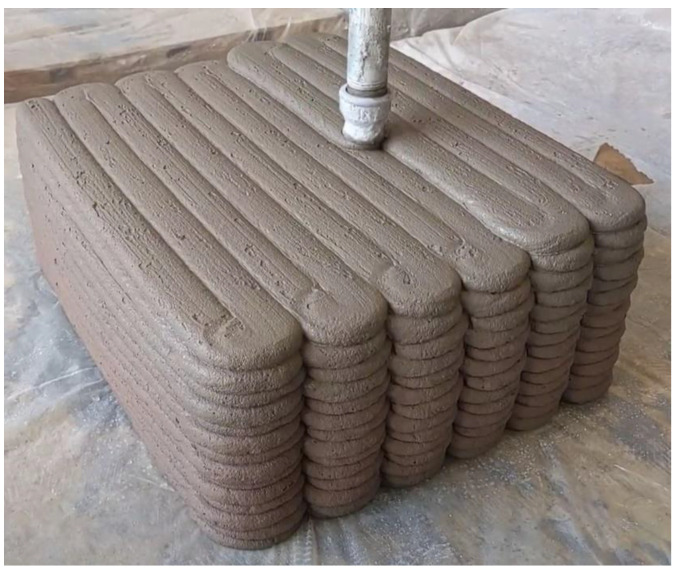
Concrete printing path.

**Figure 3 materials-16-02534-f003:**
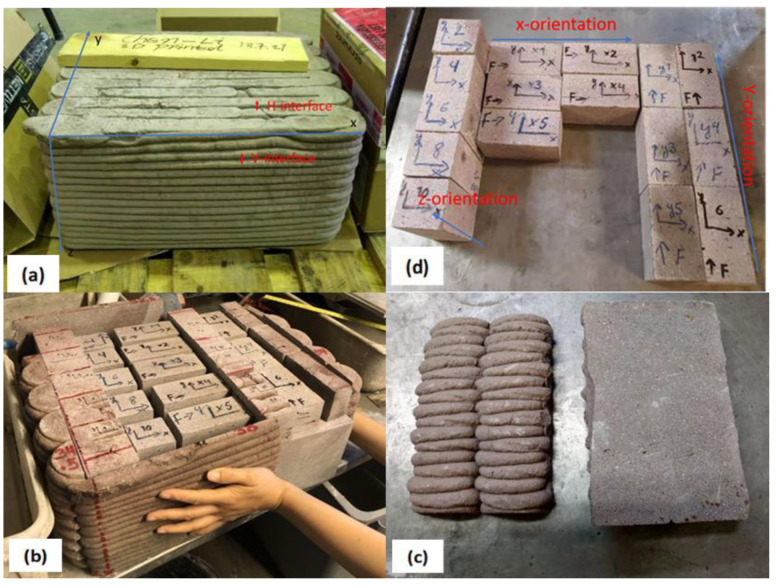
Sampling process (**a**–**d**) for printed specimens (5 cm × 5 cm × 10 cm) obtained from the printed concrete slab (38 cm × 36 cm × 22 cm).

**Figure 4 materials-16-02534-f004:**
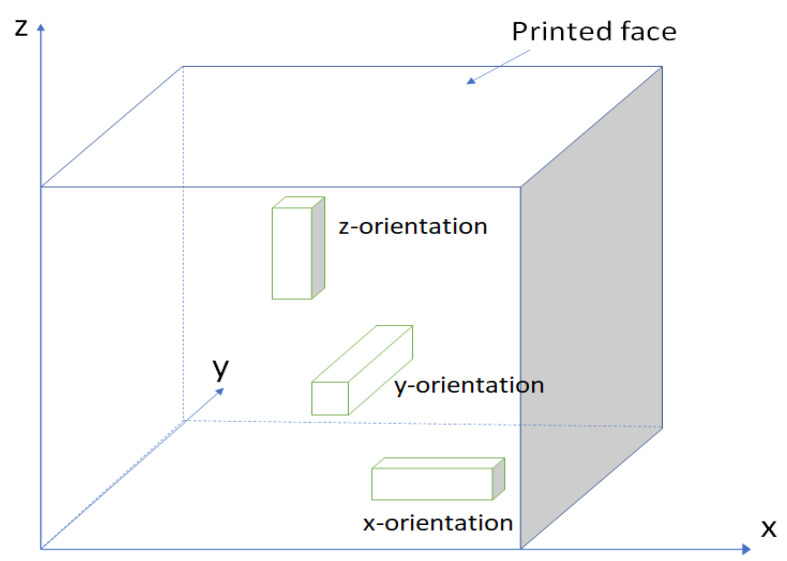
Scheme of specimen sawing.

**Figure 5 materials-16-02534-f005:**
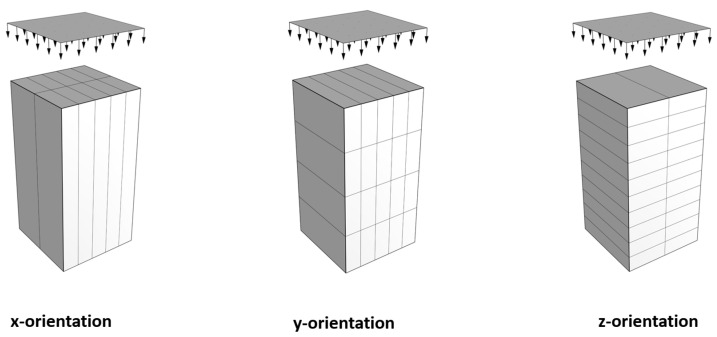
Interfaces distributions for printed specimens.

**Figure 6 materials-16-02534-f006:**
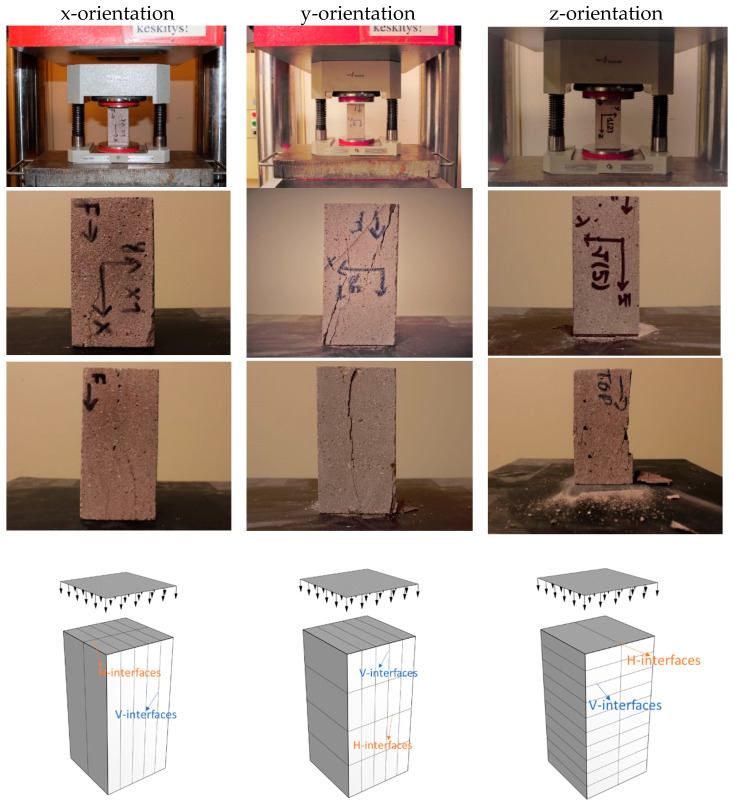
Compression test setup and failure patters for printed prisms.

**Figure 7 materials-16-02534-f007:**
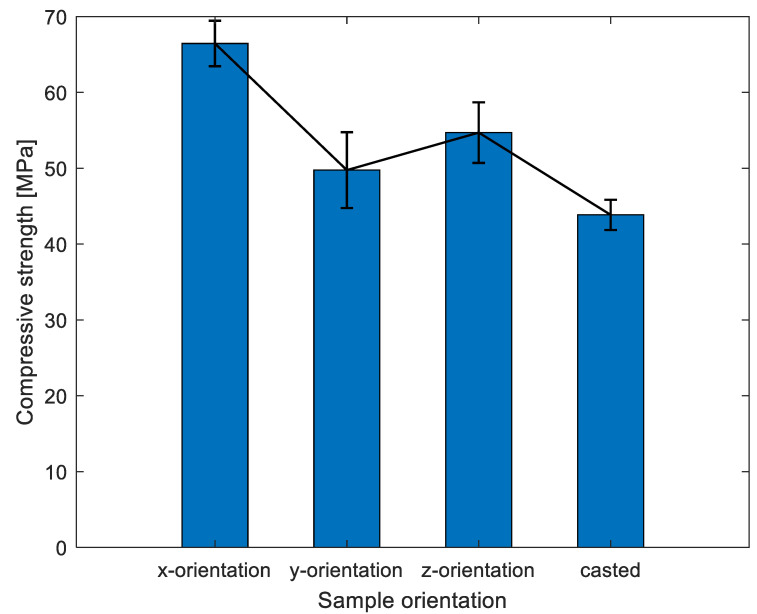
Compressive strength test results of different orientations. Notations: results in each group obtain the average from five paralleled samples.

**Figure 8 materials-16-02534-f008:**
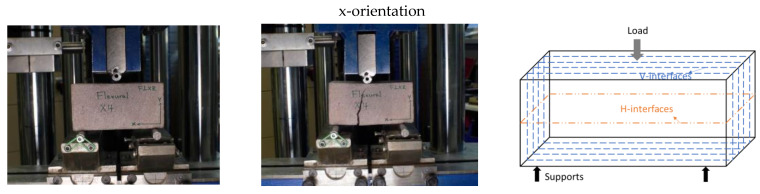
Flexural setup and failure patters for printed prisms. Note: there is no force diagram for casted prism because casted one has no interfaces.

**Figure 9 materials-16-02534-f009:**
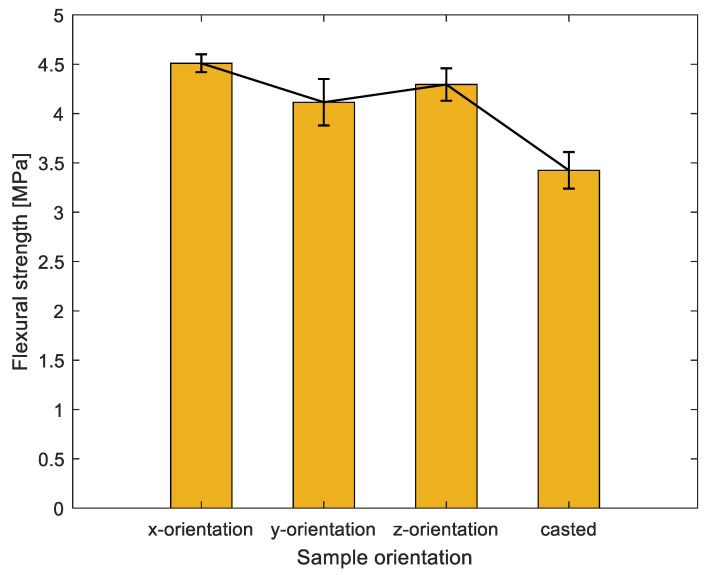
Flexural strength results of different orientations.

**Figure 10 materials-16-02534-f010:**
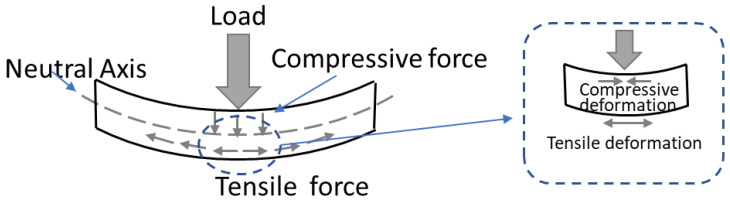
Force diagram under flexural loading.

**Figure 11 materials-16-02534-f011:**
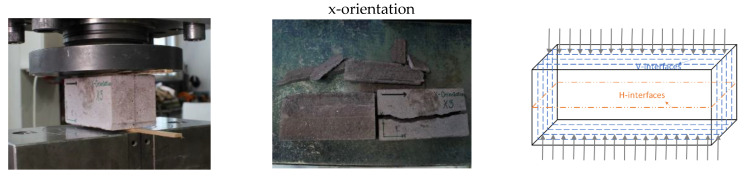
Splitting tension setup and failure patterns for printed prisms.

**Figure 12 materials-16-02534-f012:**
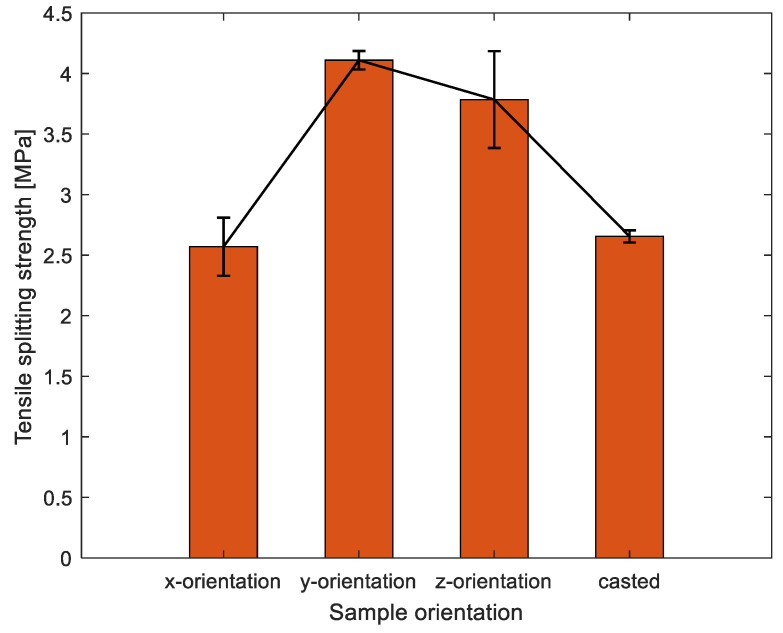
Tensile splitting strength results of different orientations.

**Figure 13 materials-16-02534-f013:**
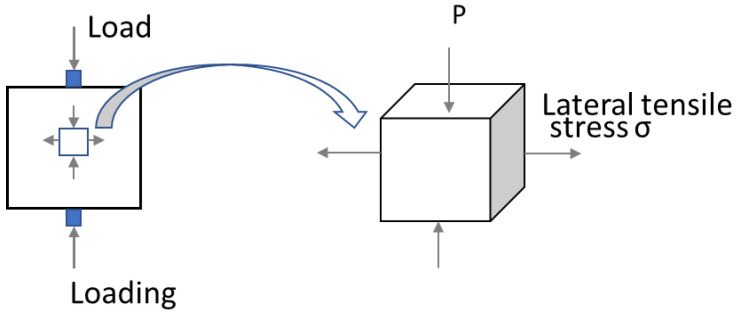
Force diagram under splitting tensile loading.

**Figure 14 materials-16-02534-f014:**
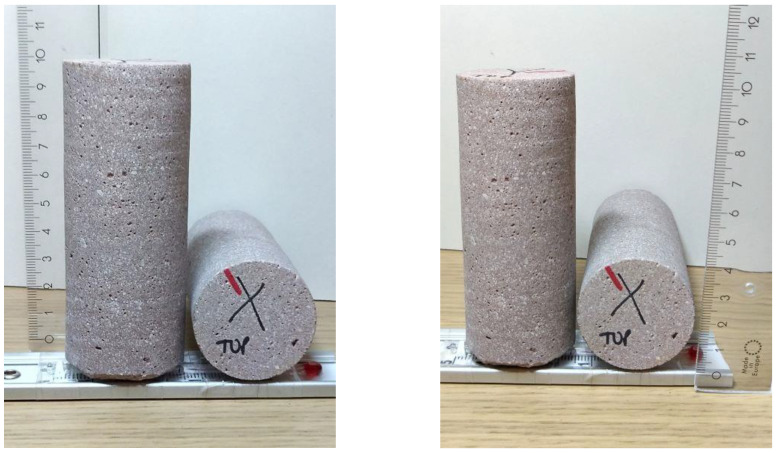
Samples drilled from the 3D-printed concrete used for X-CT tests.

**Figure 15 materials-16-02534-f015:**
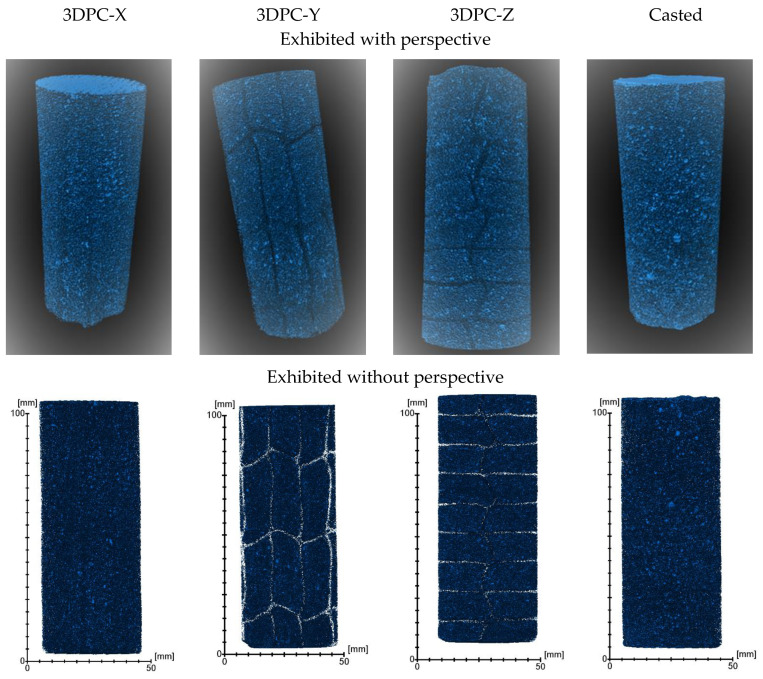
Three-dimensional rending of inter-structure from the X-CT data for samples.

**Figure 16 materials-16-02534-f016:**
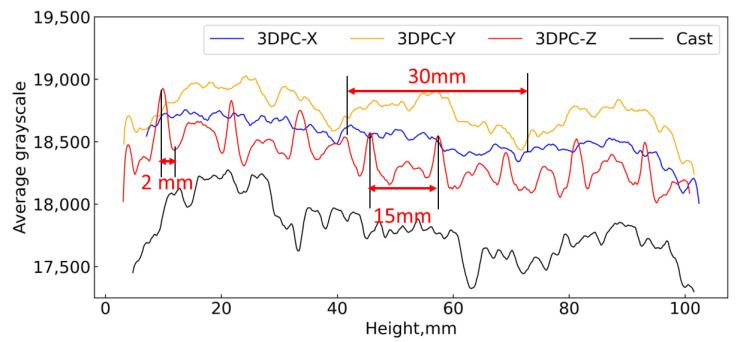
Mean greyscale values related with the height of samples.

**Figure 17 materials-16-02534-f017:**
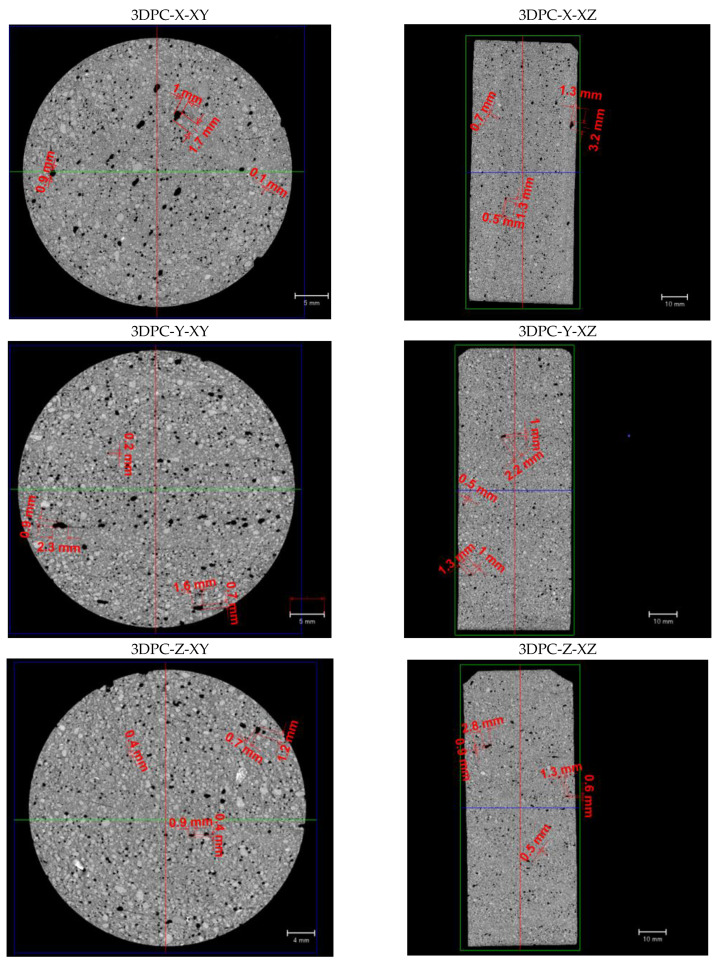
Two-dimensional CT scanning slice images of different samples: Horizonal cross-section (circle shape) and longitudinal section (rectangular shape) from the X-CT data.

**Figure 18 materials-16-02534-f018:**
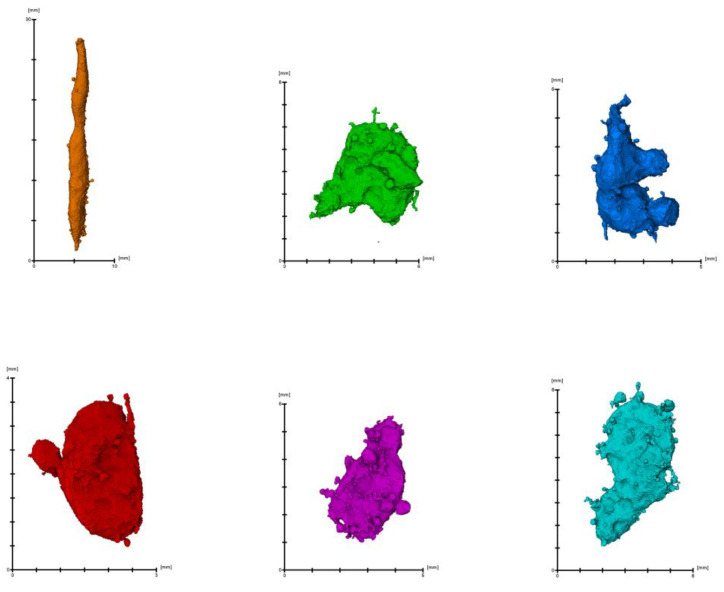
Visualized shapes of pores in 3DPC.

**Figure 19 materials-16-02534-f019:**
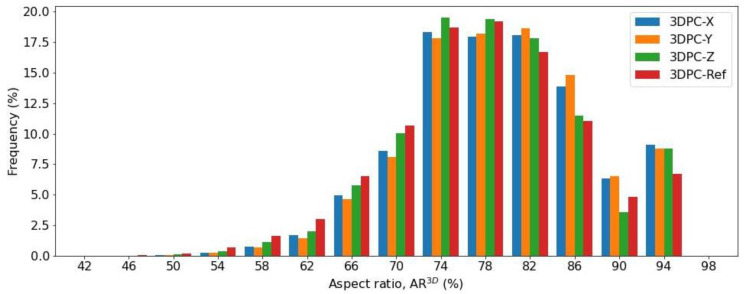
Statistical analysis of probability distribution of the 3D aspect ratio for voids in concretes.

**Figure 20 materials-16-02534-f020:**
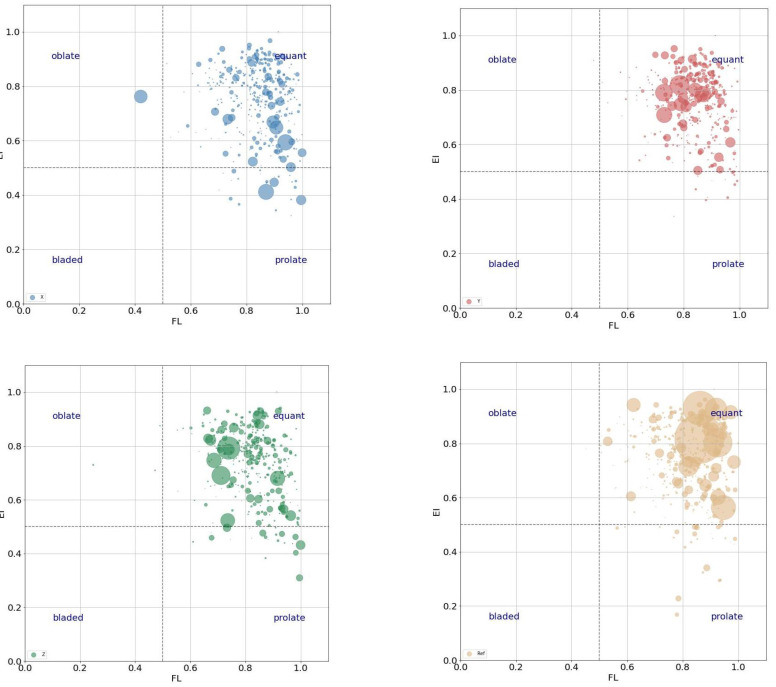
Relationship between elongation, flatness, and volume (bubble size) of the particular voids in relation to the pore shapes distinguished.

**Figure 21 materials-16-02534-f021:**
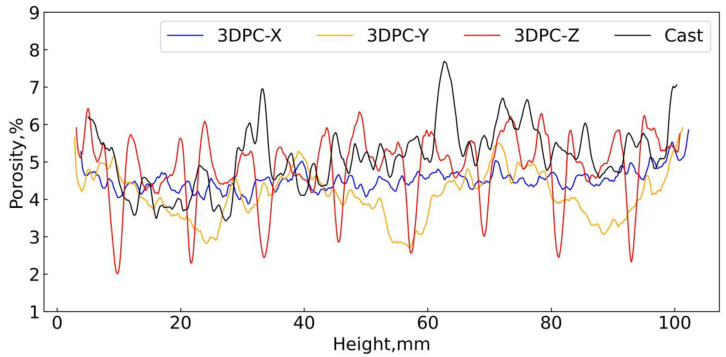
Profile of the 3D porosity with the height of samples.

**Figure 22 materials-16-02534-f022:**
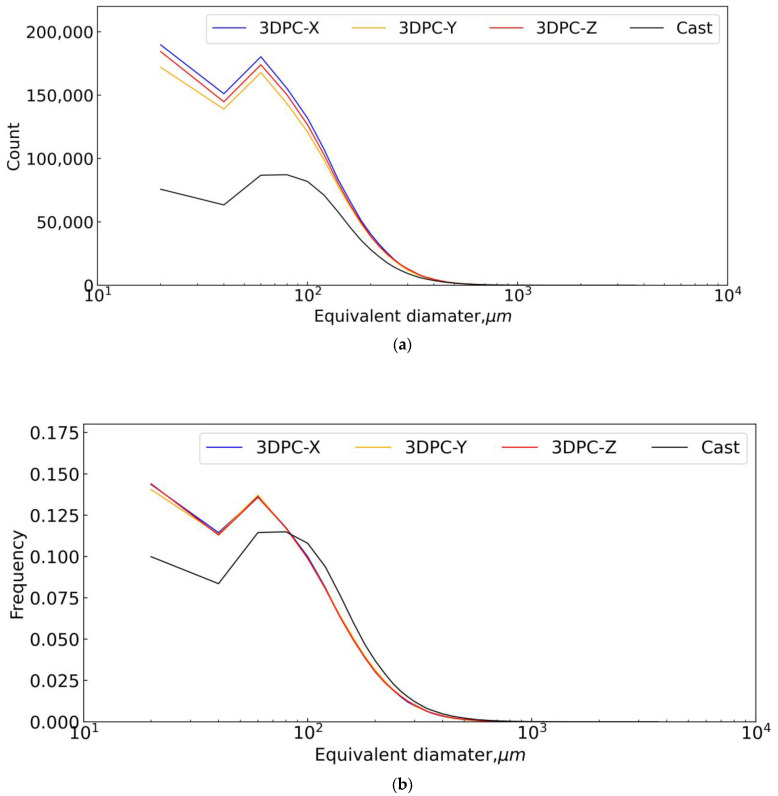
Pore size distribution of three-dimensional reconstruction to pore structure of concrete specimens. (**a**) counts, (**b**) normalized frequency.

**Table 1 materials-16-02534-t001:** Concrete quality for different values of UPV [41].

Pulse Velocity (km/s)	Concrete Quality	Rating
>4.0	Very good	1
3.5 to 4.0	Good, but may be porous	
3.0 to 3.5	Poor	2
2.5 to 3.0	Very poor	3
2.0 to 2.5	Very poor and low integrity	
<2.0 and reading fluctuating	No integrity, large voids suspected	4

**Table 2 materials-16-02534-t002:** Results of the UPV and the correlated compressive strength.

NO.	Pulse Velocity (km/s)	Correlated Strength (MPa)	Tested Strength (MPa)	Variation between Correlated and Tested Strength
x-orientation	4.15	57.8	66.4	13%
y-orientation	3.88	54.5	49.8	9%
z-orientation	3.9	54.8	54.7	2%
casted	3.6	51.4	43.9	17%

Notation: The value of pulse velocity in each group obtained the mean value from at least three paralleled specimens.

**Table 3 materials-16-02534-t003:** Pore information of concretes from X-CT scanning.

	Total Porosity (%)			Number of Slices	Number of Pores
	Minimum	Maximum	Average		
3DPC-X	3.9	5.7	4.5	3932	>104 M
3DPC-Y	2.7	5.8	4.1	3936	>104 M
3DPC-Z	2.0	6.4	4.8	3908	>104 M
Casted	3.4	7.5	5.1	3810	>100 M

**Table 4 materials-16-02534-t004:** Values of distance from the adjacent valley and peak.

	Values of Distance from Valley to Peak (mm)		
	Minimum	Maximum	Average
H-interface	12.9	15.5	14.4
V-interface	1.5	2.6	1.9

**Table 5 materials-16-02534-t005:** Statistic of values of porosity for the interfaces.

	Values of Porosity from Valley to Peak (%)							
H-interface	2.8–5.1		2.7–5.4			3.0–5.8		
V-interface	2.8–5.8	2.0–5.6	2.3–5.9	2.4–5.3	2.8–6.2	2.5–5.7	3.0–6.1	2.4–5.7

## Data Availability

The data presented in this study are available on request from the corresponding author.

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
