# Peer review of "Investigation of the Internal Structure of Hardened 3D-Printed Concrete by X-CT Scanning and Its Influence on the Mechanical Performance"

_materials, 2023, doi:10.3390/ma16062534_

Round 1
Reviewer 1 Report
The authors presented complex work about the characterization of 3D printed concrete microstructure and mechanical performance. The obtained data have been compared with concrete prepared in the traditional way. Before final acceptance, I have a few points that need to be improved:
1. please check the whole manuscript for typos like CO2 without number written using subscript
2. Fig. 3 – I miss a more detailed explanation of the presented collection of images – please indicate inserted four images as a-d and insert a clear description
3. 2.7 – here should be all relevant info about the set-up of mCT measurements (not in the Result and discussion part – pages 13-14) and also a description add how did you calculate quantitative parameters from measured mCT volumes
4. Page 13 – part 4. – remove identical sentences: X-ray . . . . .(the same is written in Exp. part)
5. Part 4.1. – as written above, move it to exp. part
6. Drilled samples – did you detect using mCT some cracks/microcracks inside these samples formed due to the drilling procedure?
7. Part 4.3. – in my opinion, it will be better to present calculated total porosity values reported in Table 3 and then go to the details like shapes of pores, probability distr. (Fig. 19), . . . .
8. Fig. 18 – better description is needed
9. As written above, I miss details on how all values describing the porosity have been calculated from mCT volumes (specify used software, procedures, . . . .)
10. Fig. 20 – there are two figures marked as 20 (see pages 20 and 21)
11. Fig. 20 (page 20) – improved the description of image and increase the size of inserted legends – it is almost not visible
12. Page 18 – clarify the sentences started as The observed irregular . . . . .
Author Response
Thanks for taking the time to review our manuscript. We appreciate all your input. Enclosed is our response to every point. Please see the attachment.

Reviewer 2 Report
The results of “Investigation of the internal structure of hardened 3D printed concrete by X-CT scanning and its influence on the mechanical performance.” are of potential interest. The introduction section provides sufficient background of past literatures. In the experimental Programme section, all the testing methods are sufficiently described. In the experimental result and discussion section, the results are elaborately discussed with figures and tables. The conclusions are supported by the results. All the references are related to this research and also sufficient. However, the following corrections are to be carried out before the acceptance of the Manuscript.
1. Abstract: Give the full form of 3DPC. Research need is to be included. Include the research recommendation. Present the specific results obtained in the abstract.
2. Introduction needs to be strengthened a bit.
3. Mention the novelty/research gap of your research.
4. Mention the relevant Standards/Codes followed for testing
5. Compare your results with existing literatures.
6. Table 2. Give the error percentage between correlated and tested strength.
7. What is your recommendation/future scope of your research? Present it in the conclusion section.
8. Mention your research significance/impact in the manuscript.
9. References need to formatted according to journal guidelines
Author Response
Enclosed is our point-by-point response. Please see the attachment.

Reviewer 3 Report
Dear Author,
1. The introduction should clearly explain the key limitations of prior work that are relevant to this paper.
2. Contributions should be highlighted more. It should be made clear what is novel and how it addresses the limitations of prior work.
3. The authors should explain clearly what the differences are between the prior work and the solution presented in this paper.
4. Some text must be added to discuss the future work or research opportunities
5. The discussion for experimental result can be improved.
6. In table 5, the second column words should not be in bold
7. In figure 17, many dimension details are not visible, use different colour.
Author Response

(The authors gave the same response as above.)

Round 2
Reviewer 1 Report
Manuscript was improved according to my comments, thus I suggest to accept it.